# Parameter-Efficient Fine-Tuning via Partially Decomposable Loss Analysis and Sharing

## Abstract

Large language model (LLM) has become a crucial tool for many machine learning research and applications. Due to the large parameter count of these models and the enormous amount of training data, large language models are usually strong at general tasks. For most applications, however, one would like a smaller, more parameter-efficient model that is specialized in a particular field. This motivates the design of fine-tuning, which tunes a pre-trained LLM for a few iterations on a dedicated dataset for specific tasks. If not handled correctly, the fine-tuning process would create another LLM that has a comparable amount of parameters, significantly slowing down any downstream applications.

One of the most widely-known ideas for resolving this issue is the Low-Rank Adaptation (LoRA) framework, where one assumes the fine-tuning weights are low-rank therefore the number of parameters together with the inference time is drastically improved. While performing well in practice, LoRA method is still a heuristic and lacks theoretical guarantees even though the loss function might inherit certain structures. Moreover, when fine-tuning multiple similar tasks in parallel, LoRA requires one to learn a pair of distinct low-rank matrices for each task, ignoring possible shared structure between tasks.

In this work, we design a framework that further reduces parameter count compared to LoRA and enables parameter sharing across different parallel fine-tuning tasks. When the number of parallel fine-tuning tasks grows larger, we cut the parameter count almost in half compared to LoRA. Moreover, we prove why our approach and more generally, LoRA works for a large class of loss functions. We empirically verify the effectiveness of our method on various benchmark models and datasets, demonstrating much-improved parameter count while retaining similar performance as LoRA.

## 1 Introduction

Large language models (LLMs) such as GPT-4 (OpenAI, 2023) are becoming increasingly important in the realm of artificial intelligence and information technology, serving a multitude of functions across various sectors (Chiang et al., 2023; Touvron et al., 2023a;b). Their ability to understand, generate, and interact with human language in a nuanced manner makes them invaluable tools in everything from customer service and data analysis to content creation and decision support systems. Beyond automating tasks, LLMs contribute to the development of conversational agents that can assist with mental health (Nori et al., 2023), offer educational tutoring (Nori et al., 2023; Phung et al., 2023; Kasneci et al., 2023), and provide specialized advice in legal or medical fields (Bommarito II & Katz, 2022; Wang et al., 2023). These models can process and analyze vast amounts of data far more quickly than humans, making them particularly useful in sifting through large datasets to identify trends or insights. Thus, LLMs are not only reshaping our interaction with technology but also have the potential to significantly impact how we solve complex problems, improve efficiency, and enhance the quality of life.

Fine-tuning is an essential step in harnessing the full potential of Large language models (Touvron et al., 2023b; Wang et al., 2023), tailoring their generalized capabilities to meet specific needs or goals. While these models are trained on a broad range of data to perform various tasks, they often require further customization to excel in specialized applications. Fine-tuning allows businesses,

researchers, and developers to adapt LLMs for particular industries, such as healthcare (Bommarito II & Katz, 2022; Wang et al., 2023), finance (Yang et al., 2023; Wu et al., 2023), or law (Bommarito II & Katz, 2022), thereby optimizing their performance and making them more effective and reliable tools. This customization not only improves the model's utility but also helps in mitigating biases, ensuring ethical use, and meeting compliance standards. In essence, fine-tuning is the bridge between a model's generalized abilities and its application in solving real-world, domain-specific problems, making it a critical element in the deployment of LLMs across diverse settings. Moreover, fine-tuning is crucial for commercial deployment of LLMs (OpenAI, 2023; Touvron et al., 2023a;b), as they offer a simple, lightweighted and efficient approach to perform fast inference for dedicated tasks. For example, fine-tuning LLMs has been consequential for Enterprise co-pilots (E.g., github co-pilot) that are being deployed widely.

The process of fine-tuning a Large language model involves several methods, each with its unique advantages, depending on the application and goals. One common approach is data augmentation (Shorten et al., 2021; Feng et al., 2021; Yoo et al., 2021), where the existing dataset is expanded by adding variations of the data to increase diversity and reduce overfitting. Another method is curriculum learning (Xu et al., 2020; Bengio et al., 2009), which involves progressively training the model on increasingly complex tasks, allowing it to build up its expertise gradually. Transfer learning (Chronopoulou et al., 2019; Houlsby et al., 2019) is also widely used, taking a pre-trained model and adapting it for a specific task by training it further on a specialized dataset. Feature-based fine-tuning involves extracting certain layers or "features" from the pre-trained model and incorporating them into a new model designed for the specific task. Hyperparameter tuning, where settings like learning rate or batch size are adjusted, is also crucial for optimizing performance. Additionally, multi-task learning (Sanh et al., 2021; Liu et al., 2019a) can be employed to fine-tune the model on several related tasks simultaneously, thereby enhancing its generalizability. These methods can be used individually or in combination to ensure that the model performs optimally in its designated role, making fine-tuning a versatile and indispensable step in the deployment of large language models.

One of the key aspects to address is the parameter count of fine-tuning (Houlsby et al., 2019; Mangrulkar et al., 2022; Ding et al., 2023). Let $W_0 \in \mathbb{R}^{d \times m}$ denote the pre-training model weight, note that if done naively, even fine-tuning on a single data point will end up with a model as large as $W_0$, as the $\Delta W \in \mathbb{R}^{d \times m}$ is without any structure if no further assumptions are imposed. On the other hand, fine-tuning should be parameter-efficient and highly structured, as most of the technical heavy-lifting has been handled by the time- and parameter-consuming pre-trained model $W_0$. Drawing inspiration from deep learning theory, where the gradients of model weights are usually low-rank due to over-parametrization, Hu et al. (2021) proposes the Low-Rank Adaptation (LoRA) framework, where they assume the fine-tuning weights, $\Delta W$ admits a rank-$r$ factorization for a hyperparameter $r$. While $d$ and $m$ can be as large as $10^4$ to $10^5$ and lead to a pre-trained model with trillions of parameters, the LoRA approach allows one to pick $r = 50$ or $100$, reduces the parameter count by more than $100\times$ fold. Moreover, by performing the fine-tuning process on a low-rank model $\Delta W = AB$ for $A \in \mathbb{R}^{d \times r}$ and $B \in \mathbb{R}^{r \times m}$, they also effectively improve the inference time, as multiplying a vector with $\Delta W$ can be performed by first multiplying with $B$ then computing the matrix-vector product using $A$ and the resulting vector, a runtime improvement from $O(md)$ to $O(mr + dr)$. Due to these advantages, LoRA has become an important building block for the fine-tuning procedure of many LLMs, including GPT-3 (Brown et al., 2020).

Despite its impressive empirical performance, LoRA does have several drawbacks. The method itself is still a heuristic, as the work Hu et al. (2021) does not provide convergence guarantees and they instead motivate the effectiveness of LoRA from the perspective of subspace similarity Hamm & Lee (2008). From an algorithmic perspective, LoRA requires each individual fine-tuning task to learn a distinct pair of low-rank matrices $A_i, B_i$. This ignores the potential relevance between different tasks, for example, in the AEP Copilot fine-tuning pipeline, the content and citation retrieval are highly correlated, and are fine-tuned on the same LLM. One would expect the similarity and relevance of these two tasks can be exploited in a manner that further reduces the number of parameters required for fine-tuning. With these considerations in mind, we ask the following question:

*Can we develop a framework that further reduces the parameter count for parallel fine-tuning, and provide theoretical guarantees for it?*

In this work, we provide a positive answer to the above question. In particular, we propose a framework called *Multiple Parallel Low-Rank Adaptation (`Multi-LoRA`)* that enables one to fine-

tune $k$ tasks with the same pre-trained model in a more parameter-efficient manner than LoRA. Moreover, we prove that for a large class of loss functions, our method exhibits a good convergence speed under mild assumptions on the gradient. Our analysis, surprisingly, draws inspiration from the convergence analysis of FedAvg (McMahan et al., 2017) and its variant (Song et al., 2023). We also empirically verify the effectiveness of our proposed method on publicly available Roberta (Liu et al., 2019b) and GPT-2 (Radford et al., 2019) models. We achieve similar performance as LoRA while significantly reduce the parameter count. We summarize our contributions as follows:

- We propose `Multi-LoRA`, a framework that enables fine-tuning multiple tasks in parallel and reduces parameter count. Specifically, for $k$ tasks, LoRA would require $O(kdr + kmr)$ parameters, while Multi-LoRA only requires $O(dr + kmr)$ parameters. If $d$ and $m$ are in the same order and $k$ is large, this almost cuts the number of parameters in half.

- We provide theoretical guarantees for our framework via a partially decomposable loss analysis. In particular, we show that whenever the individual task loss has a good local structure (such as Lipschitzness, smoothness, and convexity), it can be propagated to the global loss.

- We empirically verify our algorithm by performing experiments on Roberta and GPT2 on various benchmarks as in Hu et al. (2021). We achieve comparable performances as Hu et al. (2021), while only using $\leq 60\%$ of parameters compared to LoRA.

## 1.1 OUR RESULTS

We start with the formulation of our loss,

**Definition 1.1** (A mathematical interpretation of the model). *Let $L : \mathbb{R}^d \times \mathbb{R}^m \to \mathbb{R}$ denote a loss function. Let $k \geq 1$ denote a positive integer. Define the global loss function $\mathcal{L} : \mathbb{R}^d \times \mathbb{R}^{mk} \to \mathbb{R}$ as follows $\mathcal{L}(x, y) := \sum_{i=1}^{k} L(x, y_i)$ where $y \in \mathbb{R}^{mk}$ is a vector that concatenates all $k$ parameters for each $y_i$: $y := \begin{bmatrix} y_1^\top & y_2^\top & \cdots & y_k^\top \end{bmatrix}^\top$.*

**Lemma 1.2** (Informal of Lemma 4.4, lipschitz). *If $L$ is $\gamma$-Lipschitz, then $\mathcal{L}$ is $(\gamma k)$-Lipschitz.*

**Lemma 1.3** (Informal of Lemma 4.6, smoothness). *If $L$ is $\beta$-smooth, then $\mathcal{L}$ is $(k\beta)$-smooth.*

Our results for Lipschitzness and smoothness don't require additional assumptions. However, for strong convexity, extra structural assumptions are needed (for details see Lemma 4.8) as otherwise counterexample exists.

**Lemma 1.4** (Informal of Lemma 4.8, strongly convex). *If $L$ is $\alpha$-strongly convex and under certain assumptions, then $\mathcal{L}$ is $(0.5\alpha)$-strongly convex.*

Consequentially, standard first-order optimization methods can be applied directly to `Multi-LoRA` and convergence can be obtained.

## 2 RELATED WORK

**Practical/Empirical LLMs**  In the landscape of practical and empirical studies on Large Language Models (LLMs), a plethora of research has emerged that scrutinizes various facets of these computational behemoths. Studies often focus on the utility of LLMs in specific sectors such as healthcare, where they have been deployed for diagnostic assistance and drug discovery, or in the financial sector for risk assessment and fraud detection. Additionally, benchmarking papers have explored the raw performance of these models, evaluating them across various natural language processing tasks like machine translation, text summarization, and question-answering. BERT (Devlin et al., 2018) presents a novel approach in NLP by pre-training deep bidirectional representations on unlabeled text, which enables fine-tuning with minimal architectural modification to achieve state-of-the-art results across various language processing tasks. Across the GPT (Radford et al., 2018; 2019; Brown et al., 2020; OpenAI, 2023) series, the progression of GPT models is evident, each showcasing advancements in natural language understanding and generative capabilities. Starting with a focus on generative pre-training to improve task-specific fine-tuning in diverse NLP tasks, the series evolves towards the development of GPT-4 (OpenAI, 2023), a multimodal, large-scale model capable of human-level performance in professional and academic benchmarks, accepting both image and text

inputs, and exhibiting advancements in zero-shot and few-shot learning, scalability, and applicability across a myriad of NLP tasks and real-world scenarios., T5 (Raffel et al., 2020) innovates in transfer learning for NLP by introducing a unified, text-to-text framework to harmonize diverse language problems and by comparing various elements like pre-training objectives and architectures. The PaLM series (Chowdhery et al., 2022; Anil et al., 2023) develops large-scale Transformer language models, with PaLM (Chowdhery et al., 2022) leveraging a novel ML system, Pathways to achieve SOTA few-shot learning and breakthrough performance in multi-step reasoning and multilingual tasks. PaLM 2 (Anil et al., 2023) builds upon this by improving multilingual capabilities, reasoning, and compute efficiency. FLAN (Wei et al., 2021)introduces a model that enhances zero-shot learning in language models through instruction tuning, a process of fine-tuning models on tasks described via instructions, showing superior performance over models like GPT-3 in various NLP tasks by utilizing natural language instructions, model scale, and diversified fine-tuning datasets. A significant body of work has also been dedicated to the fine-tuning methods that optimize LLMs for specialized tasks, offering critical insights into effective techniques such as data augmentation, transfer learning, and hyperparameter tuning. For example, RoBERTa (Liu et al., 2019b) refines BERT pretraining approach, revealing through a meticulous replication study that by optimizing key hyperparameters and training data size, BERT was significantly undertrained and could achieve state-of-the-art results on several NLP benchmarks, thus emphasizing the pivotal role of design choices in model performance. Ethical considerations, particularly those related to bias and fairness, have also seen growing empirical examination. These research efforts collectively contribute to the ongoing refinement of LLMs, enhancing their practicality and ethical standards, and ensuring their effective deployment across a wide range of applications.

**Theoretical LLMs** A number of work have studied the Large Language Models. Several work have tried to explain the LLMs in different kind of angles Panigrahi et al. (2023a); Sanford et al. (2023); Tarzanagh et al. (2023); Arora & Goyal (2023); Malladi et al. (2023b); Panigrahi et al. (2023b). There are also a number of work studied the attention scheme in computation level, such as Kitaev et al. (2020); Wang et al. (2020); Zandieh et al. (2023); Alman & Song (2023); Deng et al. (2023); Brand et al. (2023); Malladi et al. (2023a)

## 3 MULTI-LORA: A PARAMETER-SHARING FRAMEWORK FOR EFFICIENT FINE-TUNING

In this section, we present Multi-LoRA, a framework that enables multiple fine-tuning tasks to share a same low-rank module while maintaining their identities. This results in a nearly 50% parameter reduction compared to LoRA. To make the following discussion clear, for matrices $A, B$, we use subscript $A_i, B_i$ to denote the low-rank modules for the $i$-th task, and we use $A(t), B(t)$ to denote the low-rank modules for the $t$ iteration. For example, $B_i(t)$ would denote the low-rank module for the $i$-th task at iteration $t$.

At a high level, Multi-LoRA resembles that of LoRA: the algorithm tunes two low-rank modules iteratively. The key difference here is that Multi-LoRA further freezes the module $A$ and shares it across all different tasks. This approach seems to weaken the LoRA framework, as it reduces the expressivity of the model by freezing $A$ and further, if the tasks are highly unrelated, then sharing $A$ would intuitively be a bad idea. We show both theoretically and empirically that this is not the case. Theoretically, we provide a mathematical formulation of our Multi-LoRA framework and observe a partially decomposable structure of this formulation. This enables one to transmit good local structure to the global loss, therefore explaining the good performance of Multi-LoRA. Empirically, we observe Multi-LoRA exhibits comparable performance as LoRA while reducing number of parameters dramatically.

## 4 MATHEMATICAL FORMULATION OF MULTIPLE PARALLEL FINE-TUNING

In this section, we provide a mathematical model for Multi-LoRAfrom the perspective of loss functions. Specifically, we show that by our formulation of global loss, the local structure can be propagated to global.

---

**Algorithm 1** Parameter reduction through sharing a low-rank module across all fine-tuning tasks. We use $W_0$ to denote the pre-trained model weights, $r$ to denote the rank parameter of low-rank modules, $L$ to denote the loss function that takes in a pre-trained weight, a low-rank factorization of fine-tuning weight and an $m$-dimensional data point. $k$ denotes the total number of tasks to be fine-tuned, and $X_1, \ldots, X_k$ denote specific datasets for each task. Finally, let $T$ denote the total number of fine-tuning episodes.

---

1: **procedure** MULTI-LORA($W_0 \in \mathbb{R}^{d \times m}, r, L : \mathbb{R}^{d \times m} \times \mathbb{R}^{d \times r} \times \mathbb{R}^{r \times m} \times \mathbb{R}^m \to \mathbb{R}, k, X_1 \in \mathbb{R}^{m \times n_1}, \ldots, X_k \in \mathbb{R}^{m \times n_k}, T$)
2:     /* Initialization stage: initialize $A(0)$ with random Gaussian and $B_i(0)$ to $\mathbf{0}_{r \times m}$. */
3:     Initialize each entry of $A(0)$ as independent $\mathcal{N}(0, 1)$
4:     **for** $i = 1 \to k$ **do**
5:         $B_i(0) \leftarrow \mathbf{0}_{r \times m}$
6:     **end for**
7:     /* Fine-tuning stage: freeze $W_0$ and $A(0)$, while tuning for each $B_i(t)$. */
8:     **for** $t = 1 \to T$ **do**
9:         $A(t) \leftarrow A(t-1)$                          ▷ Freeze shared module $A$.
10:         **for** $i = 1 \to k$ **do**
11:             Update $B_i(t)$ using $B_i(t-1)$ and $\nabla_B L(W_0, A(t), B_i(t-1), X_i)$
12:         **end for**
13:     **end for**
14:     **return** $A(T), \{B_i(T)\}_{i=1}^k$
15: **end procedure**

---

## 4.1 NOTATIONS

We use $[n]$ for a positive integer $n$ to denote the set $\{1, 2, \ldots, n\}$. Given a vector $x$, we use $\|x\|_2$ to denote its $\ell_2$ norm. Given a matrix $A$, we use $\|A\|$ to denote its spectral norm and $\|A\|_F$ to denote its Frobenious norm. Given a matrix $A$, we use $A_{i,*}$ to denote its $i$-th row and $A_{*,j}$ to denote its $j$-th column. We use $\|A\|_0$ to denote the $\ell_0$ semi-norm of $A$ that measures the number of nonzero entries in $A$.

Given a rank-$k$, $m \times n$ real matrix $A$, we use $\kappa(A)$ to denote its condition number: $\kappa(A) = \frac{\sigma_1(A)}{\sigma_k(A)}$ where $\sigma_1(A), \ldots, \sigma_k(A)$ are singular values of $A$ sorted in magnitude. When $A$ is clear from context, we often use $\kappa$ directly.

For a $n \times n$ matrix $A$, we say it is positive semi-definite (PSD, $A \succeq 0$) if for all $x$, $x^\top A x \geq 0$. We say it is positive definite (PD, $A \succ 0$) if for all $x \in \mathbb{R}^n \backslash \mathbf{0}_n$, we have $x^\top A x > 0$.

## 4.2 ASSUMPTIONS AND DEFINITIONS

In this section, we provide a mathematical formulation of the multiple parallel fine-tuning tasks model studied in this paper.

**Definition 4.1** (A mathematical interpretation of the model, restatement of Defintion 1.1). *Let $L : \mathbb{R}^d \times \mathbb{R}^m \to \mathbb{R}$ denote a loss function. Let $k \geq 1$ denote a positive integer.*

*Define the global loss function $\mathcal{L} : \mathbb{R}^d \times \mathbb{R}^{mk} \to \mathbb{R}$ as follows*

$$\mathcal{L}(x, y) := \sum_{i=1}^k L(x, y_i)$$

*where $y \in \mathbb{R}^{mk}$ is a vector that concatenates all $k$ parameters for each $y_i$:*

$$y := \begin{bmatrix} y_1 \\ y_2 \\ \vdots \\ y_k \end{bmatrix}.$$

We would like to highlight the connection between this formulation and the fine-tuning scheme we are to introduce in this paper. Our algorithm would resemble that of LoRA, with the significant difference

that if there are $k$ tasks, LoRA would maintain $k$ pairs of matrices $A_i, B_i$, but our algorithm would *share* a single $A$ across all $k$ tasks while only let each task vary their $B_i$'s. In the above formulation, one can view $x$ as a shared parameter across all local losses and $y_i$'s are the customized parameters that are unique to each task. The global loss function is heavily inspired by federated learning and the FedAvg paradigm McMahan et al. (2017). We would also like to remark that we can vary the loss function by replacing $L$ with $L_i$ and our results still hold. For the simplicity of presentation, we stick to the same loss across all tasks.

The gradient of the loss function can be compactly expressed in the following way:

**Claim 4.2.** *We have*

$$\frac{\mathrm{d}\mathcal{L}(x,y)}{\mathrm{d}(x,y)} = \begin{bmatrix} \sum_{i=1}^{k} \frac{\mathrm{d}L(x,y_i)}{\mathrm{d}x} \\ \frac{\mathrm{d}L(x,y_1)}{\mathrm{d}y_1} \\ \vdots \\ \frac{\mathrm{d}L(x,y_k)}{\mathrm{d}y_k} \end{bmatrix}$$

*Proof.* The proof of this statement follows straightforward by using chain rule. □

The form of the gradient enables us to perform a "partial decomposition": we can decompose it into a term contributed by the shared term, and $k$ terms correspond to individual parameters. This motivates us to prove structural property of the global loss based on individual loss. Below, we show several standard properties of loss functions, such as Lipschitzness, smoothness and (strong) convexity can be readily propagated from individual loss to global loss.

### 4.3 Lipschitz property

**Definition 4.3.** *We say function $L : \mathbb{R}^d \times \mathbb{R}^m \to \mathbb{R}$ is $\gamma$-Lipschitz if*

$$|L(x,y_i) - L(\widetilde{x},\widetilde{y}_i)| \leq \gamma \cdot \| \begin{bmatrix} x \\ y_i \end{bmatrix} - \begin{bmatrix} \widetilde{x} \\ \widetilde{y}_i \end{bmatrix} \|_2$$

**Lemma 4.4** (Formal version of Lemma 1.2). *If $L : \mathbb{R}^d \times \mathbb{R}^m \to \mathbb{R}$ is $\gamma$-Lipschitz, then $\mathcal{L} : \mathbb{R}^d \times \mathbb{R}^{mk} \to \mathbb{R}$ is $(\gamma k)$-Lipschitz.*

*Proof.* We can show

$$|\mathcal{L}(x,y) - \mathcal{L}(\widetilde{x},\widetilde{y})|^2 = |\sum_{i=1}^{k} (L(x,y_i) - L(\widetilde{x},\widetilde{y}_i))|^2$$

$$\leq k \sum_{i=1}^{k} |L(x,y_i) - L(\widetilde{x},\widetilde{y}_i)|^2$$

$$\leq k \sum_{i=1}^{k} \gamma^2 \| \begin{bmatrix} x \\ y_i \end{bmatrix} - \begin{bmatrix} \widetilde{x} \\ \widetilde{y}_i \end{bmatrix} \|_2^2$$

$$= k\gamma^2 \sum_{i=1}^{k} (\|x - \widetilde{x}\|_2^2 + \|y_i - \widetilde{y}_i\|_2^2)$$

$$\leq k^2\gamma^2 (\|x - \widetilde{x}\|_2^2 + \sum_{i=1}^{k} \|y_i - \widetilde{y}_i\|_2^2)$$

$$= k^2\gamma^2 (\|x - \widetilde{x}\|_2^2 + \|y - \widetilde{y}\|_2^2)$$

$$= k^2\gamma^2 \cdot \| \begin{bmatrix} x \\ y \end{bmatrix} - \begin{bmatrix} \widetilde{x} \\ \widetilde{y} \end{bmatrix} \|_2^2$$

where the first step follows from definition of $\mathcal{L}$, the second step follows from Cauchy-Schwarz inequality, the third step follows from loss function $L$ is $\gamma$-Lipschitz, the forth step follows definition of $\ell_2$ norm, the fifth step follows from simple algebra, the sixth step follows from definition of $y$.

Taking the square root of both sides of the above equation, then we have

$$|\mathcal{L}(x,y) - \mathcal{L}(\widetilde{x},\widetilde{y})| \leq k\gamma \cdot \| \begin{bmatrix} x \\ y \end{bmatrix} - \begin{bmatrix} \widetilde{x} \\ \widetilde{y} \end{bmatrix} \|_2$$

Thus, we complete the proof. □

### 4.4 SMOOTHNESS

**Definition 4.5.** *We say function $L : \mathbb{R}^d \times \mathbb{R}^m \to \mathbb{R}$ is $\beta$-smooth if*

$$\|\nabla L(x,y_i) - \nabla L(\widetilde{x},\widetilde{y}_i)\|_2 \leq \beta \cdot \| \begin{bmatrix} x \\ y_i \end{bmatrix} - \begin{bmatrix} \widetilde{x} \\ \widetilde{y}_i \end{bmatrix} \|_2$$

*Here $\nabla L(x,y_i)$ denote the gradient.*

**Lemma 4.6** (Formal version of Lemma 1.3)**.** *If function $L$ is $\beta$-smooth, then $\mathcal{L}$ is $(k\beta)$-smooth.*

*Proof.* We can show

$$\begin{aligned}
\|\nabla \mathcal{L}(x,y) - \nabla \mathcal{L}(\widetilde{x},\widetilde{y})\|_2^2 &= \| \sum_{i=1}^k (\nabla L(x,y_i) - \nabla L(\widetilde{x},\widetilde{y}_i))\|_2^2 \\
&\leq k \sum_{i=1}^k \|\nabla L(x,y_i) - \nabla L(\widetilde{x},\widetilde{y}_i)\|_2^2 \\
&\leq k \sum_{i=1}^k \beta^2 \| \begin{bmatrix} x \\ y_i \end{bmatrix} - \begin{bmatrix} \widetilde{x} \\ \widetilde{y}_i \end{bmatrix} \|_2^2 \\
&= k\beta^2 \sum_{i=1}^k (\|x - \widetilde{x}\|_2^2 + \|y_i - \widetilde{y}_i\|_2^2) \\
&\leq k^2 \beta^2 (\|x - \widetilde{x}\|_2^2 + \sum_{i=1}^k \|y_i - \widetilde{y}_i\|_2^2) \\
&= k^2 \beta^2 (\|x - \widetilde{x}\|_2^2 + \|y - \widetilde{y}\|_2^2) \\
&= k^2 \beta^2 \cdot \| \begin{bmatrix} x \\ y \end{bmatrix} - \begin{bmatrix} \widetilde{x} \\ \widetilde{y} \end{bmatrix} \|_2^2
\end{aligned}$$

where the first step follows from definition of $\mathcal{L}$, the second step follows from Cauchy-Schwarz inequality, the third step follows from loss function $L$ is $\beta$-smooth, the forth step follows definition of $\ell_2$ norm, the fifth step follows from simple algebra, the sixth step follows from definition of $y$.

Taking the square root of both sides of the above equation, then we have

$$\|\nabla \mathcal{L}(x,y) - \nabla \mathcal{L}(\widetilde{x},\widetilde{y})\|_2 \leq k\beta \cdot \| \begin{bmatrix} x \\ y \end{bmatrix} - \begin{bmatrix} \widetilde{x} \\ \widetilde{y} \end{bmatrix} \|_2$$

Thus, we complete the proof. □

### 4.5 CONVEXITY

**Definition 4.7.** *We say function $L : \mathbb{R}^d \times \mathbb{R}^m \to \mathbb{R}$ is $\alpha$-strongly convex if*

$$\nabla^2 L(x,y_i) \succeq \alpha \cdot I_{d+m}$$

Without any assumption on $\alpha_0, \alpha, k$ relationship, the hessian $\nabla^2 \mathcal{L}(x,y)$ might not be psd due to certain counter-example (see Section A).

**Lemma 4.8** (Formal version of Lemma 1.4)**.** *If the following conditions hold*

- *Let $k \geq 1$ denote a positive integer.*

- $\nabla_x^2 L(x, y_i) \preceq \alpha_0 \cdot I_d$, where $\alpha_0 > 0$.

- $L : \mathbb{R}^d \times \mathbb{R}^m \to \mathbb{R}$ is $\alpha$-strongly convex, where $\alpha > 0$.

- Let $(k-1)(\alpha_0 - \alpha) \leq \alpha/2$.

Then we have $\mathcal{L} : \mathbb{R}^d \times \mathbb{R}^{km} \to \mathbb{R}$ is $\alpha/2$-strongly convex.

*Proof.* Note that

$$\nabla^2 \mathcal{L}(x, y) \in \mathbb{R}^{(d+km) \times (d+km)}$$

We can consider the Hessian matrix as $(k+1) \times (k+1)$ blocks.

For convenient, let us rewrite $\nabla^2 \mathcal{L}(x, y)$ as follows

$$\nabla^2 \mathcal{L}(x, y) = \begin{bmatrix} H_{0,0} & H_{0,1} & \cdots & H_{0,k} \\ H_{1,0} & H_{1,1} & \cdots & H_{1,k} \\ \vdots & \vdots & \ddots & \vdots \\ H_{k,0} & H_{k,1} & \cdots & H_{k,k} \end{bmatrix}$$

where $H_{0,0} \in \mathbb{R}^{d \times d}$, $H_{i,i} \in \mathbb{R}^{m \times m}$ for all $i \in [k]$, $H_{0,i} \in \mathbb{R}^{m \times d}$, $H_{i,0} \in \mathbb{R}^{d \times m}$ and $H_{i,j} \in \mathbb{R}^{m \times m}$. Since $y_i$ and $y_j$ has no correlation in loss function, thus $H_{i,j} \in \mathbf{0}_{m \times m}$ is an all zero matrix.

From assumption we know $L$ is $\alpha$-strongly convex, thus for all $i \in [k]$, we know

$$\begin{bmatrix} H_{0,0} & H_{0,i} \\ H_{i,0} & H_{i,i} \end{bmatrix} \succeq \alpha \cdot I_{m+d}$$

Let us pick up a vector $z \in \mathbb{R}^{d+mk}$, let $z_0 \in \mathbb{R}^d$, let $z_i \in \mathbb{R}^m$ for all $i \in [k]$. Then we have

$$\begin{aligned}
z^\top \nabla^2 \mathcal{L}(x, y) z &= z_0^\top H_{0,0} z_0 + \sum_{i=1}^k z_i^\top H_{i,i} z_i + \sum_{i=1}^k z_i^\top H_{i,0} z_0 + z_0^\top H_{0,i} z_i^\top \\
&= k \cdot z_0^\top H_{0,0} z_0 + \sum_{i=1}^k z_i^\top H_{i,i} z_i + \sum_{i=1}^k z_i^\top H_{i,0} z_0 + z_0^\top H_{0,i} z_i^\top - (k-1) \cdot z_0^\top H_{0,0} z_0 \\
&= \sum_{i=1}^k (z_0^\top H_{0,0} z_0 + z_i^\top H_{i,i} z_i + z_i^\top H_{i,0} z_0 + z_0^\top H_{0,i} z_i) - (k-1) \cdot z_0^\top H_{0,0} z_0 \\
&\geq \sum_{i=1}^k \alpha(\|z_0\|_2^2 + \|z_i\|_2^2) - (k-1) z_0^\top H_{0,0} z_0 \\
&\geq \sum_{i=1}^k \alpha(\|z_0\|_2^2 + \|z_i\|_2^2) - (k-1)\alpha_0 \|z_0\|_2^2 \\
&= \alpha \|z\|_2^2 - (k-1) \cdot (\alpha_0 - \alpha) \|z_0\|_2^2 \\
&\geq \alpha \|z\|_2^2 - (k-1) \cdot (\alpha_0 - \alpha) \|z\|_2^2 \\
&\geq \alpha \|z\|_2^2 - 0.5\alpha \|z\|_2^2 \\
&= 0.5\alpha \|z\|_2^2
\end{aligned}$$

where the first follows from definition of Hessian, the second step follows from simple algebra, the third step follows from simple algebra, the forth step follows from $L$ is $\alpha$-strongly convex, the fifth step follows from $\nabla_x^2 L(x, y_i) \preceq \alpha_0 \cdot I_d$, the sixth step follows from simple algebra, the seventh step follows from $\|z_0\|_2^2 \leq \|z\|_2^2$, the eight step follows from assumption of $\alpha_0$ and $\alpha$, the ninth step follows from simple algebra. □

Table 1: Performance of RoBERTa$_{base}$ with various adaptation techniques on the GLUE benchmark. For MNLI we provide the overall (matched and mismatched) accuracy, for CoLA we present Matthew's correlation, and for STS-B we report Pearson correlation. For the remaining tasks we disclose the accuracy. For all these metrics, a higher value is preferable. Numbers marked with * are referenced from previously published works. The best in **bold** and the second best underlined.

| Method | # of Trainable Parameters | MNLI | SST-2 | MRPC | CoLA | QNLI | QQP | RTE | STS-B | Avg. |
|---|---|---|---|---|---|---|---|---|---|---|
| Pre-trained | 0M | $33.5_{\pm 1.3}$ | $50.5_{\pm .8}$ | $61.0_{\pm 16.5}$ | $0.0_{\pm .0}$ | $50.3_{\pm .4}$ | $42.1_{\pm 11.8}$ | $48.4_{\pm 2.4}$ | $-2.2_{\pm 5.9}$ | 35.5 |
| Fine-Tuning* | 1000.0M | **87.6** | 94.8 | **90.2** | **63.6** | 92.8 | **91.9** | 78.7 | 91.2 | 86.4 |
| LoRA* | 2.4M | $87.5_{\pm .3}$ | $\textbf{95.1}_{\pm .2}$ | $89.7_{\pm .7}$ | $63.4_{\pm 1.2}$ | $\textbf{93.3}_{\pm .3}$ | $90.8_{\pm .1}$ | $\textbf{86.6}_{\pm .7}$ | $\textbf{91.5}_{\pm .2}$ | **87.2** |
| Multi-LoRA | 1.3M | $86.1_{\pm .2}$ | $94.7_{\pm .4}$ | $88.4_{\pm .7}$ | $57.9_{\pm .8}$ | $92.0_{\pm .2}$ | $89.6_{\pm .1}$ | $81.8_{\pm 2.2}$ | $90.5_{\pm .3}$ | 85.1 |

Table 2: BLEU score (↑) of GPT-2 medium (M) with various adaptation techniques on the E2E NLG Challenge, DART and WebNLG. Numbers marked with * are referenced from previously published works. The best numbers are in **bold** and the second best ones are underlined.

| Method | # of Trainable Parameters | E2E NLG Challenge | DART | WebNLG | Avg. |
|---|---|---|---|---|---|
| Fine-Tuning* | 1065M | 68.2 | 46.2 | 46.5 | 53.6 |
| LoRA | 1.1M | **68.9** | **46.2** | **54.9** | **56.7** |
| Multi-LoRA | 0.7M | 67.3 | 42.2 | 51.6 | 53.7 |

## 5 EXPERIMENTS

**RoBERTa on NLU.** We conduct experiments on the base model of RoBERTa (Liu et al., 2019b) for natural language understanding (NLU) tasks. RoBERTa refined the original pre-training methodology set forth in BERT (Devlin et al., 2018), enhancing its performance on various tasks without significantly increasing the number of parameters. We assess the effectiveness of various efficiency-focused adaptation methods on GLUE tasks using RoBERTa base (125M) from the HuggingFace Transformers library. Additionally, we replicate the configurations specified in studies by Houlsby et al. (2019). To maintain an equitable comparison, we alter two key elements in our evaluation approach for LoRA. Firstly, we maintain a consistent batch size across all tasks and fix the sequence length at 128. Secondly, the model is initialized using pre-trained settings for MRPC, RTE, and STS-B, as opposed to starting with a model pre-adapted to MNLI like in the fine-tuning baseline. To implement our algorithm, we utilize the same LoRA$_A$ layers (randomly initialized by default configuration) for all the 8 tasks and only tune LoRA$_B$ layers on each task respectively. We set the dimension of the low-rank matrices $r = 8$ and the scaling factor for the weight matrices LoRA$_\alpha = 16$ for all the experiments. the maximum sequence length is chosen as 512, the warmup ratio is selected as 0.06.

We compare our method with pre-trained RoBERTa$_{base}$ model, RoBERTa$_{base}$ fully fine-tuned, and RoBERTa$_{base}$ fine-tuned with LoRA. Since the results of Fine-Tuning and LoRA are borrowed from literature and run 5 times independently, we also run our experiments (our method and the evaluation of Pre-trained model) 5 times using random seed 0, 1, 2, 3, 4. As shown in Tab. 1, our proposed Multi-LoRA can achieve comparable results with state-of-the-art fine-tuning approaches with significantly smaller parameter size in total.

**GPT-2 on NLG.** Having demonstrated Multi-LoRA's effectiveness in comprehensive fine-tuning on NLU, we aim to investigate whether Multi-LoRA maintains its superiority on NLG (Natural Language Generation) tasks. We conduct experiments of GPT-2 medium models on E2E NLG Challenge, WebNLG and DART. We set the dimension of the low-rank matrices $r = 4$, the scaling factor for the weight matrices LoRA$_\alpha = 32$ and dropout probability of the LoRA layers as 0.1 for all the experiments. The maximum sequence length is chosen as 512. As shown in Tab. 2, our Multi-LoRA is able to maintain high NLG performance with extremely small parameter size and is better than fully fine-tuning which training parameter size is thousands of times more than ours.

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

APPENDIX

## A  COUNTEREXAMPLE FOR STRONG CONVEXITY

The following counterexample indicates that we need extra assumption to make sure that $\mathcal{L}$ has strongly convex property.

**Lemma A.1** (Counterexample). *Let $H \in \mathbb{R}^{(k+1) \times (k+1)}$ denote the following matrix*

$$
H = \begin{bmatrix}
4 & -3 & -3 & -3 & \cdots & -3 \\
-3 & 4 & 0 & 0 & \cdots & 0 \\
-3 & 0 & 4 & 0 & \cdots & 0 \\
-3 & 0 & 0 & 4 & \cdots & 0 \\
\vdots & \vdots & \vdots & \vdots & \ddots & \vdots \\
-3 & 0 & 0 & 0 & \cdots & 4
\end{bmatrix}
$$

*We know that*

- **Part 1.** $\langle H, \mathbf{1}_{(k+1) \times (k+1)} \rangle = -2k + 4$

- **Part 2.** $\begin{bmatrix} 4 & -3 \\ -3 & 4 \end{bmatrix} \succeq 0$

*Proof.* **Proof of Part 1.**

We can show

$$
\begin{aligned}
\langle H, \mathbf{1}_{(k+1) \times (k+1)} \rangle &= 4(k+1) - 6k \\
&= -2k + 4
\end{aligned}
$$

**Proof of Part 2.**

We have

$$
\begin{aligned}
\begin{bmatrix} x & y \end{bmatrix} \begin{bmatrix} 4 & -3 \\ -3 & 4 \end{bmatrix} \begin{bmatrix} x \\ y \end{bmatrix} &= 4x^2 + 4y^2 - 6xy \\
&= x^2 + y^2 + 3(x^2 - 2xy + y^2) \\
&= x^2 + y^2 + 3(x - y)^2 \\
&\geq 0
\end{aligned}
$$

$\square$

## B  SOFTMAX REGRESSION: A CASE STUDY

As a toy example, we consider the softmax regression problem Deng et al. (2023). This problem can be viewed as a simplification of two-layer transformer network. To better understand how effectiveness our method would be, we empirically verify various parameters for this problem.

**Definition B.1** (Softmax regression). *Given $A \in \mathbb{R}^{n \times d}$ and $b \in \mathbb{R}^n$, we define loss function $L : \mathbb{R}^d \to \mathbb{R}$ as*

$$
L(x) := 0.5 \| \langle \exp(Ax), \mathbf{1}_n \rangle^{-1} \exp(Ax) - b \|_2^2
$$

*Here*

- $\exp(Ax)_i = \exp((Ax)_i)$ *for all $i \in [n]$*

- $\mathbf{1}_n \in \mathbb{R}^n$ *denote the length-$n$ vector where all the entries are ones*

- $\langle a, b \rangle = \sum_{i=1}^n a_i b_i$.

**Definition B.2.** *Let $f(x) = \langle \exp(Ax), \mathbf{1}_n \rangle^{-1} \exp(Ax)$. Let $c(x) = f(x) - b$.*

**Lemma B.3** (see Definition 5.7 in page 14 in Deng et al. (2023)). *Let $g(x) \in \mathbb{R}^d$ denote the gradient of softmax regression loss (see Definition B.1), then we have*

$$g(x) = A^\top (-f(x)\langle c(x), f(x) \rangle + \mathrm{diag}(f(x))c(x))$$

**Lemma B.4** (see Definition 6.1 in page 28 in Deng et al. (2023)). *Let $H(x) \in \mathbb{R}^{d \times d}$ denote the Hessian of softmax regression loss (see Definition B.1), we have*

$$H(x) = \underbrace{A^\top}_{d \times n} \underbrace{B(x)}_{n \times n} \underbrace{A}_{n \times d}$$

*where*

$$\begin{aligned} B(x) := & \langle 3f(x) - 2b, f(x) \rangle f(x)f(x)^\top \\ & + (b \circ f(x))f(x)^\top + f(x)(b \circ f(x))^\top \\ & + \langle f(x) - b, f(x) \rangle \cdot \mathrm{diag}(f(x)) \\ & + \mathrm{diag}((2f(x) - b) \circ f(x)) \end{aligned}$$

*Here*

- $(a \circ b)_i = a_i b_i$ *for all $i \in [n]$*

**Remark B.5.** *Note that $B(x)$ is a constructed as three rank-1 matrices with two diagonal matrices.*

- Function is $\alpha$-Strongly convex

$$H(x) \succeq \alpha \cdot I_d$$

- Function is $\beta$-Smoothness (this is equivalent to gradient of function is $\beta$-Lipschitz)

$$\|H(x)\| \leq \beta \iff \|g(x) - g(y)\|_2 \leq \beta \cdot \|x - y\|_2$$

- Hessian of Function is $M$-Lipschitz

$$\|H(x) - H(y)\| \leq M \cdot \|x - y\|_2$$

- Function is $\gamma$-Lipschitz (this is equivalent to gradient of function is bounded)

$$\|L(x) - L(y)\|_2 \leq \gamma \cdot \|x - y\|_2 \iff \|g(x)\|_2 \leq \gamma$$

Table 3: Numerical estimations of softmax regression problem for various choices of $(n, d)$. We perform these experiments by randomly sampling point $x$ and constructing the Hessian matrix, then performing estimations. We report the average over 10000 random samples.

| $n$ | $d$ | $\alpha$ | $\beta$ | $M$ | $\gamma$ |
|---|---|---|---|---|---|
| $n = 10$ | $d = 5$ | 0.00 | 148.03 | 26.83 | 0.60 |
| $n = 10$ | $d = 10$ | 0.00 | 272.52 | 23.65 | 0.74 |
| $n = 10$ | $d = 20$ | 0.00 | 540.98 | 21.49 | 0.89 |
| $n = 20$ | $d = 5$ | 0.00 | 447.95 | 74.38 | 0.54 |
| $n = 20$ | $d = 10$ | 0.00 | 361.00 | 26.47 | 0.77 |
| $n = 20$ | $d = 20$ | 0.00 | 1309.38 | 52.23 | 0.91 |
| $n = 20$ | $d = 40$ | 0.00 | 2511.67 | 58.29 | 1.24 |

