# OpenReview forum: "Parameter-Efficient Fine-Tuning via Partially Decomposable Loss Analysis and Sharing"
_ICLR.cc/2024/Conference — ICLR 2024 Conference Withdrawn Submission_

### Official Review · Reviewer_rWs7 · 2023-10-29

**Soundness:** 1 poor
**Presentation:** 3 good
**Contribution:** 2 fair
**Rating:** 3
**Confidence:** 4

**Summary:**

This paper proposes a framework named multi-lora, which enables parameter sharing across different parallel fine-tuning tasks. The authors provide some theory to justify the rationality of this framework.

**Strengths:**

1. The paper is well-written and easy to read.

2. The idea that parameter sharing or parameter transferring between tasks under the framework of PEFT is valid and interesting.

**Weaknesses:**

1. The motivation of the studied problem is unclear. I doubt if there are any scenarios in reality where we need to parallel fine-tune LLMs on multiple tasks. Also, from the experimental results, we can save 50% of the parameters (~1M) at the cost of sacrificing performance compared to LoRA. Is this worth it?

2. Some important related works are missed. More related works should be discussed, such as parameter-efficient fine-tuning and multi-task learning.

3. The experiments are inadequate.
   - (1)This paper does not include necessary factors such as ablation studies and sensitivity analysis to demonstrate the effectiveness of the method.
   - (2)This paper does not compare the training time cost of different methods, which may be an important factor in the problem scenario proposed by the authors.
   - (3)Sequentially fine-tuning the tasks with parameter(LoRA_A) sharing should also be studied and compared with the proposed method thoroughly.

4. There are no conclusions.

5. I am unaware of the relationship between the theories and the effectiveness of the method. This paper uses a lot of space to prove some simple properties. Why do properties of global loss (lipschitz, smooth, convexity) explain the performance of multi-lora?

6. Why did you freeze LoRA_A layers and not fine-tune them?

7. Some statements about LoRA are wrong, such as “LoRA improves inference time”.

**Questions:**

Please see the weakness part. No additional questions.

---

### Official Review · Reviewer_Va7k · 2023-10-31

**Soundness:** 3 good
**Presentation:** 3 good
**Contribution:** 2 fair
**Rating:** 5
**Confidence:** 3

**Summary:**

The paper presents a model, Multi-LoRA, aimed at decreasing the quantity of trainable parameters when employing LoRA for parallel fine-tuning. Specifically, all tasks share a global and fixed parameter A, along with a trainable task-specific parameter B. This strategy significantly reduces the number of trainable parameters. In the proposed method, the parameter count for k tasks can be reduced from O(Kdr+kmr) to O(dr+kmr). The authors provide theoretical assurances for model convergence. Empirical experiments are performed on Roberta and GPT2 for natural language understanding and generation tasks, respectively.

**Strengths:**

1. This paper tackles a compelling problem: reducing the number of trainable parameters for parallel fine-tuning. The proposed Multi-LoRA method technically sounds. The authors provide a detailed theoretical proof of model convergence. The empirical results underscore the effectiveness of this method. Notably, the number of trainable parameters decreases from 2.4M to 1.3M for eight Natural Language Understanding (NLU) tasks and from 1.1M to 0.7M for three generative tasks.
2. The methodology is straightforward, and the experimental settings are detailed.

**Weaknesses:**

1. The performance of Multi-LoRA is not as strong as that of LoRA. For example, the average scores drop from 87.2 to 85.1 for understanding tasks and from 56.7 to 53.7 for generation tasks. While parameter reduction is significant, performance is often a more critical factor.

2. The paper does not include a comparison with multi-task learning (MTL). Both settings involve training multiple tasks simultaneously, yet no MTL methods, such as AdapterFusion[1], are compared in the experiments.

3. The paper lacks experiments on recent Large Language Models (LLMs) like Llama2 and does not provide an analysis of convergence speed.


[1]. AdapterFusion: Non-Destructive Task Composition for Transfer Learning

**Questions:**

1. How would the model’s performance be affected if parameter A were allowed to be trainable? Could this modification potentially enhance the model’s performance?

2. Are the RoBERTa and GPT models trained using fp32 precision? If so, what would be the impact on the models if mixed-precision training, such as fp16, were used? This question is particularly relevant given that recent Large Language Models (LLMs) commonly employ fp16 precision for training.

---

### Official Review · Reviewer_GVCH · 2023-11-03

**Soundness:** 2 fair
**Presentation:** 3 good
**Contribution:** 3 good
**Rating:** 3
**Confidence:** 2

**Summary:**

This paper designs a framework that reduces the parameter count even more than LoRA, in addition to enabling parameter sharing among various parallel fine-tuning tasks. When the volume of parallel fine-tuning tasks increases, the framework slashes the parameter count by nearly half in comparison to LoRA. Additionally, the authors provide theoretical evidence explaining the effectiveness of this approach—and, by extension, that of LoRA—for a wide array of loss functions. The effectiveness of the proposed method is empirically confirmed on multiple benchmark models and datasets, showcasing a substantial decrease in parameter count while maintaining performance comparable to that of LoRA.

**Strengths:**

Originality: The Multi-LoRA framework is a novel approach to fine-tuning LLMs that takes into account shared structure between tasks, which is an important consideration in many real-world applications.

Quality: the theoretical analysis of the method is also well-presented and provides insights into the properties of the method.

Clarity: the paper is well-organized and easy to follow.

Significance: The Multi-LoRA framework and the proposed method have the potential to improve the efficiency and effectiveness of fine-tuning LLMs, which is an important consideration for many real-world applications.

**Weaknesses:**

This paper can be significantly improved by more thorough experiments in several aspects:
1) More LLM with larger size
2) More metrics beyond GLUE for LLM
3) The proposed Multi-LoRA seems to achieve inferior performance for all the tasks. In this way, it is hard to validate the inefficiency of the proposed method

**Questions:**

see weaknesses